# High-resolution tracking of hyrax social interactions highlights nighttime drivers of animal sociality

Camille N. M. Bordes [1], Rosanne Beukeboom[1], Yael Goll[2], Lee Koren[1] & Amiyaal Ilany [1✉]

Network structure is a key driver of animal fitness, pathogen transmission, information spread, and population demographics in the wild. Although a considerable body of research applied network analysis to animal societies, only little effort has been devoted to separate daytime and nighttime sociality and explicitly test working hypotheses on social structures emerging at night. Here, we investigated the nighttime sociality of a wild population of rock hyraxes (*Procavia capensis*) and its relation to daytime social structure. We recorded nearly 15,000 encounters over 27 consecutive days and nights using proximity loggers. Overall, we show that hyraxes are more selective of their social affiliates at night compared to daytime. We also show that hyraxes maintain their overall network topology while reallocating the weights of social relationships at the daily and monthly scales, which could help hyraxes maintain their social structure over long periods while adapting to local constraints and generate complex social dynamics. These results suggest that complex network dynamics can be a by-product of simple daily social tactics and do not require high cognitive abilities. Our work sheds light on the function of nighttime social interactions in diurnal social species.

[1] Faculty of Life Sciences, Bar-Ilan University, Ramat Gan, Israel. [2] School of Zoology, Tel Aviv University, Tel Aviv, Israel. ✉email: amiyaal@gmail.com

Because animals are highly vulnerable when asleep[1], they must find appropriate sleeping sites to protect themselves from predators[2], implying that sleeping strategies and related behaviours are adaptive[3,4]. Social sleeping increases the chances to detect predators, helps mitigate low temperatures, and, depending on individual social status, can improve sleep quality[5,6]. For instance, social sleepers naturally synchronise their sleep[7] and spend more time in deep sleep stages than solitary individuals, which results in shorter total sleeping time[8], and shorter exposure to predators.

Sleeping in groups also exposes individuals to intra-specific aggression, but it is a lesser risk than being predated while asleep. Hence, most diurnal social species maintain sociality at night to limit predation risk[2] despite the cost of social stress. Several species of primates form larger social groups at night than during the day[9,10] and become more tolerant of conspecifics' proximity when sleeping in dangerous habitats[11–13]. When the risk of predation is higher than the risk of intra-specific aggression at night, sleeping groups become larger, denser, and less selective[14]. Conversely, when the risk of predation becomes negligible compared to the risk of being attacked by a conspecific, daytime groups either split into sub-units, sometimes leading individuals to sleep alone[15], or adapt their sleeping phases. For example, unfamiliar macaques synchronise their wakefulness more than individuals coming from the same natal group[16], which reduces the risk of intra-specific aggression from unfamiliar individuals.

Risks of predation and intra-specific aggression are further mitigated by the need for efficient thermoregulation when asleep, as well as the accessibility of sleeping sites. Indeed, the size of nighttime aggregations is limited by sleeping site availability and results in intra-specific competition for the most valuable positions[17]. In habitats where shelters are a limiting resource, animal societies have developed fission-fusion dynamics where large foraging aggregations split into smaller sleeping units to accommodate limited shelter space[15]. Under challenging thermal conditions, however, sleeping aggregations become larger to maintain body temperature[18], promoting less selective social bonds. This suggests that the choice of social partners before sleeping periods has important fitness consequences.

Despite the importance of nighttime ecology[19,20], little attention has been dedicated to explicitly quantify the social networks of diurnal species at night and test hypotheses on their active social behaviours before sleeping bouts. Such bias may be the result of decades of technical limitations in behavioural sciences. In the early stages of animal behavioural ecology, data describing animal sociality were collected via direct behavioural observations, which are spatially and temporally constrained by observers' abilities. Consequently, studies on the sociality of wild animals have mostly been limited to diurnal species (easier to observe) when observations were possible (mainly during daytime and in open areas). The recent revolution of automated data collection has increased the accuracy, resolution, and spatio-temporal range of behavioural data, facilitating the tracking of social interactions around the clock[21,22]. These advances allowed, for example, the study of co-roosting and co-denning behaviours in several species[23,24]. Yet, due to remaining difficulties in animal handling and ethical restrictions, some taxa are still subjected to a substantial bias towards their daytime social behaviour. Additionally, among biologging-based studies, most investigated the structure of animal social networks by pooling daytime and nighttime social contacts together[25,26], thus overlooking the social processes occurring around sleeping bouts. Few studies did separate social contacts between daytime and nighttime and even fewer specifically tested hypotheses about social networks of diurnal species at night (but see[27,28]). This gap is important to address considering the effect of social sleeping on animal

sleeping ecology, the importance of sleep for individual fitness[3], and the growing concern around diurnal species nocturnality under anthropogenic disturbances[29].

In this study, we use proximity biologging data and social network analysis to investigate the nighttime sociality of a wild population of rock hyraxes (*Procavia capensis*). Rock hyraxes are medium-sized mammals living in groups of 20 individuals on average. These groups usually include one resident male, several adult females, and their offspring. Hyraxes raise their young collectively, sometimes forming heterospecific groups[30], and are organised in egalitarian societies[31] following the principle of 'structural balance'[32]—meaning that affiliative social bonds tend to form between individuals connected to a common social partner.

Mainly active during daytime, they retreat into underground natural cavities at night to protect themselves from predators. As daylight lasts approximately 14 h in summer at our study site, they can spend up to 10 h a day underground. Laboratory-based studies showed that rock hyraxes sleep on average 6 to 7 h per day and that their sleep state durations are unaffected by light or dark conditions[33]. Considering that captive animals sleep longer than their wild conspecifics[8] due to lower exposure to stressful environmental conditions[4], wild rock hyraxes are likely active at night, although no study has yet determined the range of behaviours they express underground. Consequently, they are a good candidate species to explore nighttime sociality in a mainly diurnal species and how it relates to daytime social structure.

Rock hyraxes reproduce once a year during a mating season lasting a few weeks. At this occasion, groups interact with each other and with bachelor males roaming their territories, and it is not rare to observe males and females visiting other groups in search of mating partners. We took advantage of this period of heightened social activity in summer 2017 and tracked the social contacts among 28 wild hyraxes from the Ein Gedi Nature Reserve (Israel) for 27 consecutive days (Fig. 1). We intended to

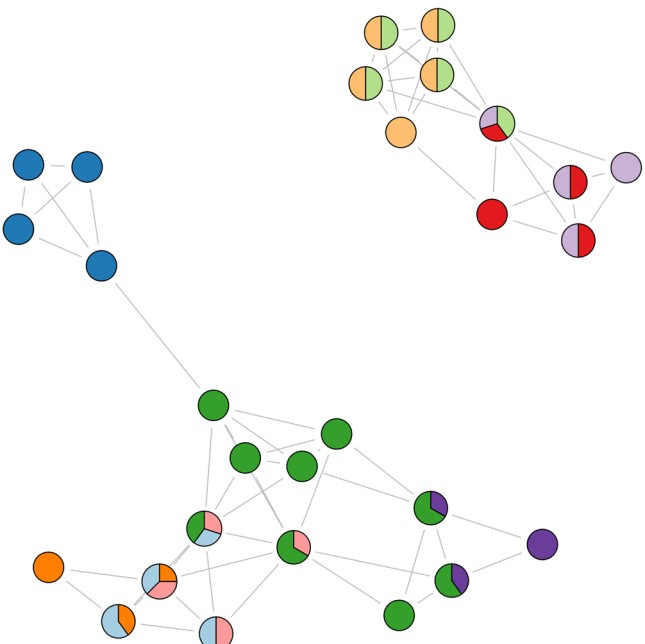

**Fig. 1 Proximity-based social network of 28 rock hyraxes aggregated over the full study period (27 days).** Circles depict individual hyraxes. Colours indicate community membership based on a community-detection algorithm. If a circle is filled with a pie chart, this individual was detected to be a member of multiple communities, and the pie charts represent the proportion of social interactions it allocates to its different communities.

(1) characterise their social behaviour at night, (2) determine if their nighttime social structure can predict daytime social structure, and (3) describe social changes occurring over crepuscule. Although hyraxes are not strictly diurnal, we expected them to sleep mainly at night, underground. Thus, we should observe more contacts during daytime but longer encounters during nighttime. As hyraxes mainly interact within their social group, with whom they share a common den at night and most of their daytime activities, we expect group composition to remain constant between day and night. Due to negligible predation risk underground, we predict animals will be more selective when foraging during the day.

We show that hyraxes readjust their social interactions before sleeping. They were found to be less social and consistently more selective of their social partners at night, supporting the idea that potential sleeping partners are carefully chosen. We suggest that nighttime sociality may have strong impact on social bonds expressed in other contexts. At the monthly scale, hyraxes maintain the binary structure of their network while reallocating their social interactions between the members of their group. This pattern likely helps hyraxes to maintain long-term social bonds while adapting to short-term changes in their physical environment. Our findings shed light on social network dynamics at a short timescale and strengthen the idea that studying the social structure of diurnal species at night advances our understanding of their ecology.

## Results

**Temporal distribution of hyrax interactions**. Raw encounter duration ranged from 11 to 25,605 s (~6 h), with 95% of all recorded contacts being shorter than 6576 s (1.8 h), and 70% shorter than 1512 s (~25 min). Hyraxes interacted more during daytime (one-sided Student t-test for dependent samples: $t = 12.734$, df = 27, $p < 0.001$, mean difference [95% CI] = 73.18 [61,39; 84.97]). On average, we recorded $n = 62.32$ (±sd = 19.41) social encounters per night and $n = 135.50$ (±31.72) encounters per day. Daytime interactions were shorter on average than nighttime interactions (mean daytime interaction: 393.40 (±659.66) seconds; mean nighttime interaction: 793.93 (±1508.75) seconds; one-sided Wilcoxon rank test for dependent samples: $V = 406$, $p < 0.001$).

Sleep is associated with lower levels of awareness[1], which affects individuals' likelihood to initiate interactions or end existing ones. When two awake individuals are engaged in a long interaction, they may break the ongoing encounter at any moment. But once animals are asleep, the contact lasts as long as both individuals remain unconscious. Consequently, social encounters recorded when two individuals are asleep are not the result of a repeated and active choice to remain near each other ('active' contacts). Rather, they are the result of a social behaviour expressed while awake and being carried out after losing consciousness ('passive' contacts). Due to their length, 'passive' contacts strongly affect the social structure of an aggregated network, which may mask the 'active' sociality expressed in-between sleeping bouts. Since we assume that nighttime social structure is predominantly sleep-related in rock hyraxes, comparing social behaviours between daytime and nighttime requires we discriminate between both types of contacts. A preliminary analysis (see Supplementary Methods) showed that a threshold of 25 min in interaction length accurately discriminates between two different social structures, prompting us to divide contacts into 'passive' (>25 min) and 'active' (<=25 min) in the rest of this study (Supplementary Fig. 1). Overall, 44.6% of nighttime proximity events were labelled as 'passive' vs. 20.1% during daytime. 'Passive' social encounters

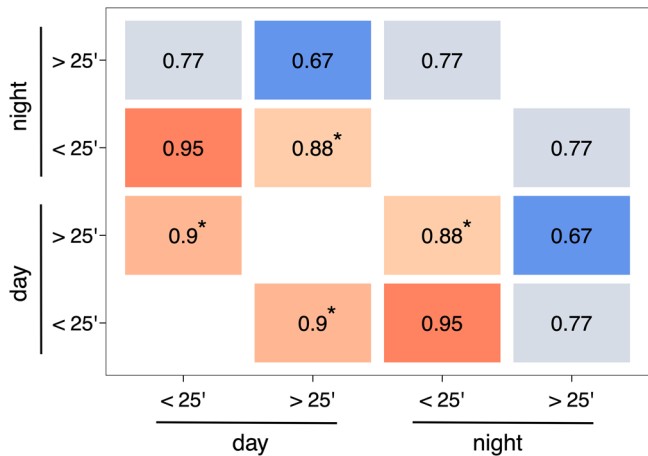

**Fig. 2 Correlation matrix between social networks built on different times of the day (night or day) and across social contexts (interactions longer or shorter than 25 min).** '*'The permutation test is significant at the level of 0.05.

accounted for 30.2% of daytime and nighttime hyrax sociality combined.

**Daytime and nighttime social structure across social contexts.** Because of differences in behavioural states, we expected 'passive nighttime' networks to be relatively poor predictors of any other type of network (i.e. 'active nighttime', 'active daytime', 'passive daytime'). As environmental conditions are different between daytime and nighttime, we also expected 'active daytime' networks to poorly predict 'active nighttime' networks. Yet, since group members synchronise their activities throughout the day, we expected hyraxes to rest with individuals sharing their activities, and 'active' and 'passive' daytime networks to be correlated.

Daytime 'passive' networks were correlated with both daytime ($r^2 = 0.90$, $p = 0.01$) and nighttime 'active' networks ($r^2 = 0.88$, $p < 0.001$). Daytime and nighttime 'active' networks predicted each other well ($r^2 = 0.95$), but this result was only marginally significant according to the permutation test ($p = 0.08$). All other similarity indexes were not significant according to the permutation test (Fig. 2).

**Comparing 'active' network traits between day and night.** At the individual level, hyraxes consistently had fewer 'active' connections at night compared to daytime, but these differences were not larger than expected by chance (average daytime degree centrality ± sd: 1.594 ± 1.229, nighttime: 0.768 ± 0.953, permutation test for paired samples: $p = 1$). Hyraxes displayed lower strength centrality at night compared to daytime (daytime: 0.467 ± 0.361, nighttime: 0.359 ± 0.426, $p < 0.001$), meaning they form weaker social bonds at night. Individual eigenvector centrality was not significantly higher during nighttime or daytime (daytime: 0.561 ± 0.43, nighttime: 0.578 ± 0.47, $p = 0.999$). Hyraxes kept interacting with the same individuals between day and night (average neighbours' stability: 0.354, $p < 0.001$, Supplementary Fig. 2) but allocated their interactions more selectively at night (daytime individual selectivity: 1.763 ± 0.642, nighttime: 1.982 ± 0.592, $p = 0.038$) (Fig. 3).

Social groups were not significantly more differentiated at night (daytime: 2.486 ± 1.195, nighttime: 3.328 ± 1.48, $p = 0.619$), and edge density did not vary more than expected by chance between daytime and nighttime (daytime: 0.363 ± 0.298, nighttime: 0.181 ± 0.225, $p = 0.712$) (Fig. 4). Yet all groups displayed lower standard deviation in two individual centrality measures at

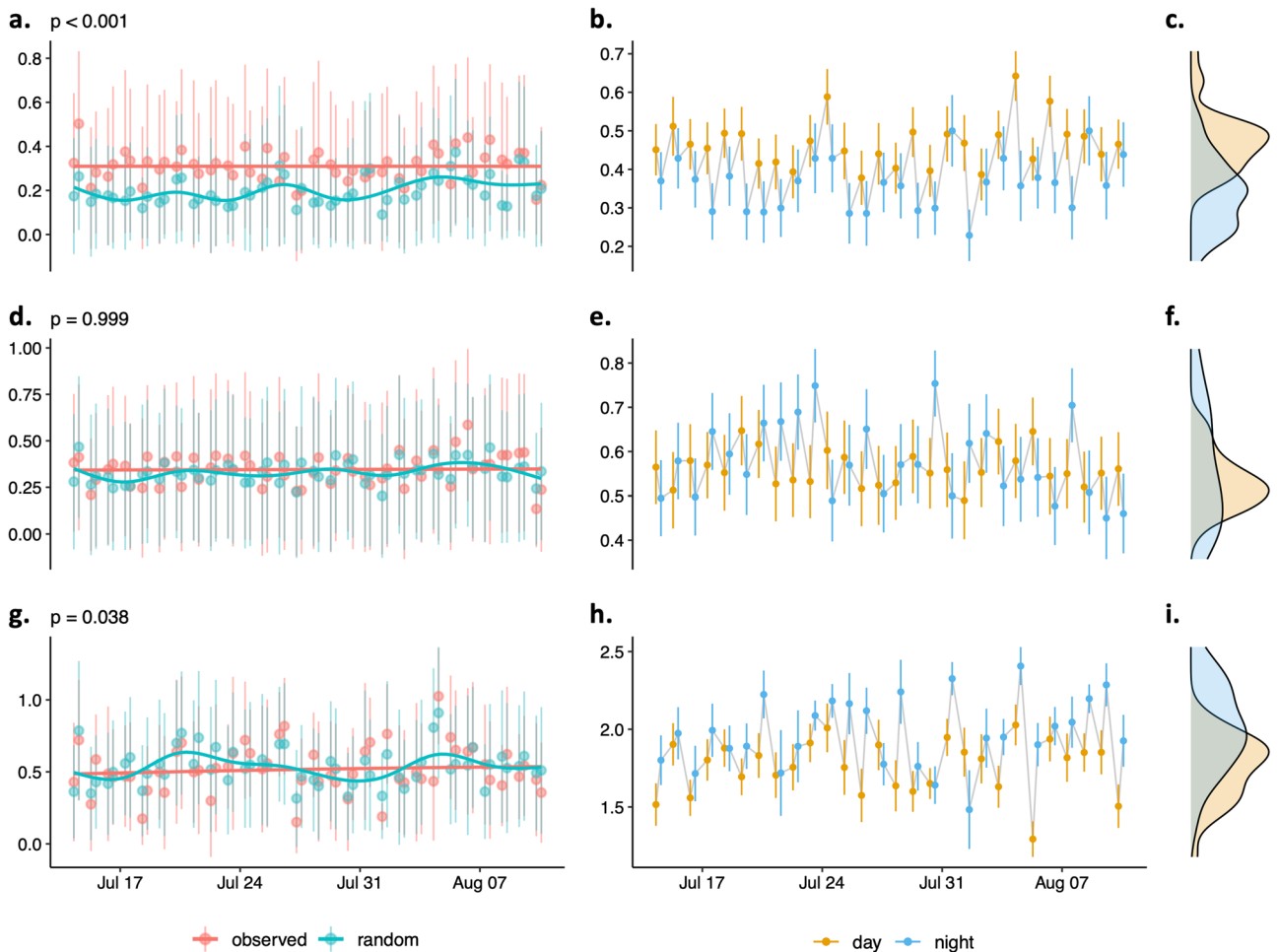

**Fig. 3 Non-random patterns in variations of network traits between consecutive days and nights.** Observed and random absolute daily differences in network traits between consecutive days and nights in (**a**) individual strength centrality, (**d**) individual eigenvector centrality, and (**g**) individual selectivity. Permutation-based *p* values determine whether the observed absolute differences between consecutive daytime and nighttime network traits were larger than random differences predicted under the null hypothesis. Average network trait per day and per night and cumulative distributions of daytime and nighttime average network traits in (**b**, **c**) individual strength centrality, (**e**, **f**) individual eigenvector centrality, and (**h**, **i**) individual selectivity. Data points represent daily averages and error bars represent the standard deviation around the mean.

night (i.e. standard deviation in strength centrality and eigenvector centrality, $p < 0.001$, Supplementary Fig. 3), meaning groups were more homogeneous at night.

**Temporal patterns of 'active' social structure**. As we assume animal space use to be the primary driver of hyrax sociality, we expected binary networks to be very similar when close in time and become less and less similar as they are further apart. Conversely, as hyraxes should re-allocate their social interactions at dawn and dusk to adjust to day-night environmental differences, we expected cosine indexes between weighted networks to be unpredictably high or low over time.

Almost all pairs of binary networks were more correlated than expected by chance. Out of 2916 pairs of networks, 2500 (85.7%) were more correlated than expected by chance, 46 (1.6%) were less correlated than expected by chance, and 534 correlations (18.3%) were not significant. Binary networks distant in time did not become less correlated than networks close in time. As expected, similarity indexes between weighted network were lower than indexes between binary network on average (weighted networks: mean $r^2 \pm sd = 0.39 \pm 0.15$, binary networks: mean $r^2 \pm sd = 0.57 \pm 0.15$; $t = 47.10^5$, $p < 0.001$). Out of 2916 pairs of networks, 830 (28.5%) were more correlated than expected by

chance, 110 (3.8%) were less correlated than expected by chance, and 2140 correlations (73.4%) were non-significant. Weighted networks showed no specific temporal patterns in the way they either correlate or diverge over time (Fig. 5).

**Discussion**

The preliminary analysis aiming at discriminating between 'active' and 'passive' social contacts confirmed that rock hyraxes are active at night and showed that 'active' social contacts accounted for up to 55.4% of nighttime proximity events. These results are consistent with previous observations of wild hyraxes venturing aboveground and being active during moonlit nights[34]. Additionally, we showed that daytime social contacts include 'passive' proximity events, which we assume to occur when animals are resting in cavities during the hottest hours of the day. Overall, our results are in favour of previous assumptions that rock hyraxes are not strictly diurnal but rather have polycyclic sleeping patterns[33].

Our results suggest that hyrax nighttime 'active' sociality drives their daytime associations. Indeed, we found that daytime and nighttime 'active' networks are highly correlated, although this level of similarity could be explained under the null hypothesis of random associations between individuals of the same group (non-

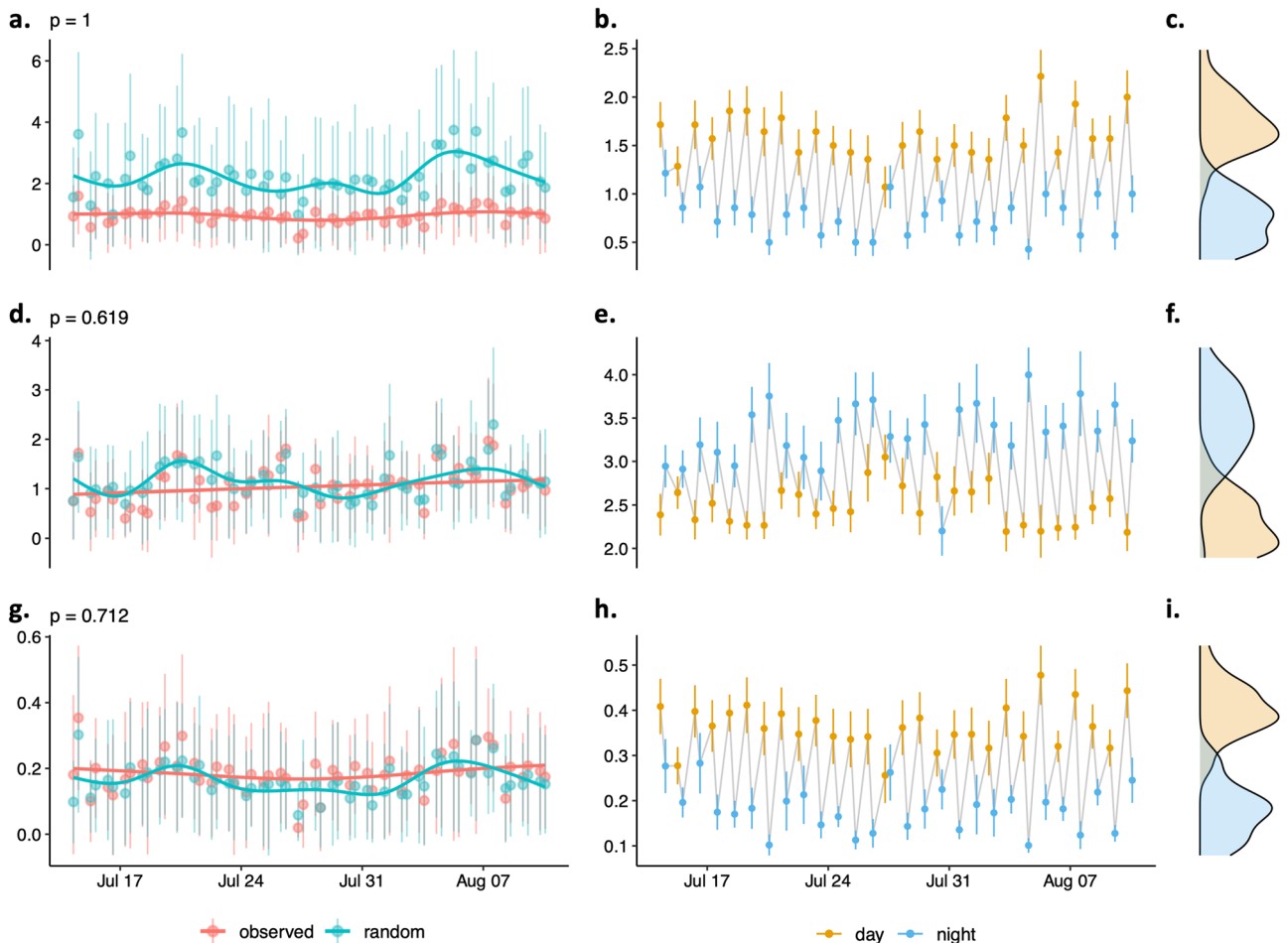

**Fig. 4 Non-random patterns in variations of network traits between consecutive days and nights.** Observed and random absolute daily differences in network traits between consecutive days and nights in (**a**) individual degree centrality, (**d**) group social differentiation, and (**g**) network density. Permutation-based *p* values determine whether the observed absolute differences between consecutive daytime and nighttime network traits were larger than random differences predicted under the null hypothesis. Average network trait per day and per night and cumulative distributions of daytime and nighttime average network traits in (**b**, **c**) individual degree centrality, (**e**, **f**) group social differentiation, and (**h**, **i**) network density. Data points represent daily averages and error bars represent the standard deviation around the mean.

significant permutation test). We also showed that hyraxes maintain the number and identity of their social partners across day and night. This suggests that rock hyraxes that interact during the day also share a sleeping den at night. Limited access to refuge drives animal movements, and consequently animal sociality, in multiple mammal species[27,35–37]. At night, den access constrains interactions between members of the same group. In the morning, hyraxes emerge from the den and forage together, rarely further than 15 metres away from a potential shelter[38]. Consequently, while foraging, we assume they favour social behaviours towards individuals who shared their den the night before. This pattern is unlikely to change outside hyrax mating season as a similar daily routine was observed in the absence of sexual competition. Hence, we suggest that spatially constrained 'active' nighttime associations generally drive hyrax 'active' daytime sociality.

Of note, the present study only highlights correlations between the different types of networks and is not sufficient to prove causality, hence we cannot exclude that daytime space use may in fact drive nighttime social networks. Nevertheless, nighttime networks constraining daytime networks is a more parsimonious interpretation of our results. Indeed, hyraxes sharing a den for the night emerge at the same place and at the same time the next morning, and then proceed to follow a leader to a common

foraging site[39]. During the mating season, animals leave their group soon after foraging activities to roam their home range in search of a mating partner but return to their group sleeping site at sunset, thus weakening the hypothesis that daytime space use constrains nighttime social interactions.

Contrary to our prediction, daytime 'passive' networks accurately predict daytime and nighttime 'active' networks, at levels beyond those predicted by hyrax space use (our null hypothesis). Several explanations can be formulated. First, hyraxes that forage together may synchronise their daytime activities as an anti-predator strategy, as seen in other species. For example, guppy shoals living in high-risk conditions display fewer fission events compared to guppies living in low-risk environments[40]. Roaming away from one's social group results in higher exposure to predators whereas staying together provides protection from threats, despite changes in behavioural activities. Second, as 'active' and 'passive' daytime social activities are adjacent in time, hyraxes may maintain their social connections because of social continuity. Indeed, they act mostly as a group: they emerge from a shared den in the morning, bask in the sun (reaching hyperthermic levels in the morning[41]), and then follow a leader to a feeding site[39] where they forage together. These activities account for most of their 'active' daytime sociality. Hyraxes later retreat to cool places where they dissipate the heat accumulated in the

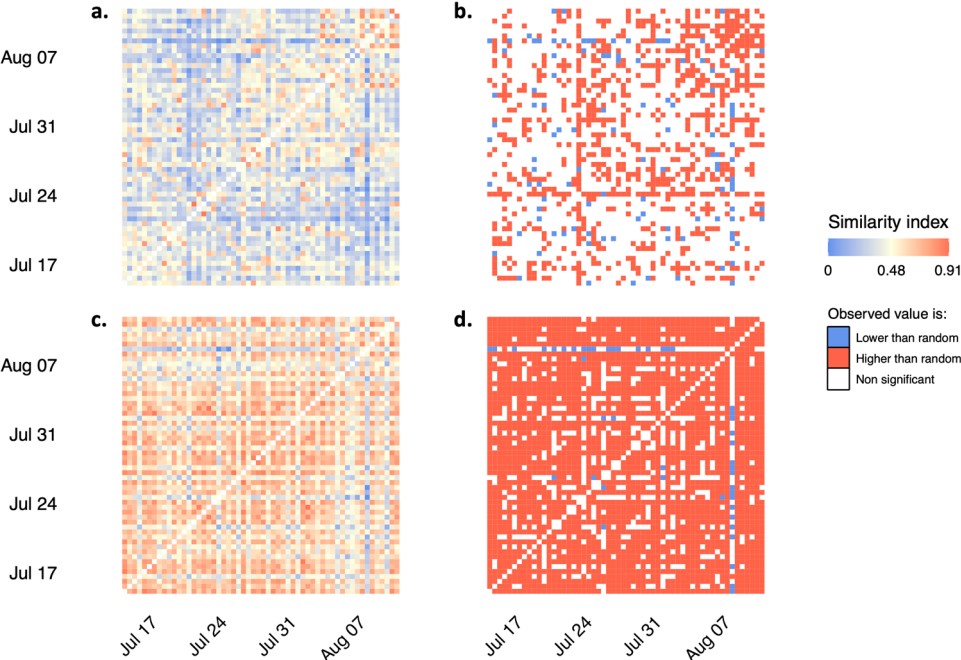

**Fig. 5 Temporal correlation between daily 'active' networks over a month. a** Pairwise cosine similarity indexes between all possible pairs of daily weighted social networks and (**b**) associated permutation-based *p* values. **c** Pairwise cosine similarity indexes between all possible pairs of daily binary social networks and (**d**) associated permutation-based *p* values.

morning via passive thermal transfer[41]. 'Passive' proximity contacts mainly occur when they thermoregulate and rest in these cavities, soon after their daily foraging activities. Thus, they maintain the same group when transitioning from morning foraging to afternoon resting. Finally, our study period covers the hyrax annual mating season[42]. Increased intra-specific competition and aggression during the mating season trigger females to stay together to reduce sexual conflicts in several species. For example, females aggregate together to dilute male sexual attention in red junglefowls[43], cockroaches[44], and mosquitofish[45]. In addition, territorial resident males drive male competitors away[46] and guard sexually receptive females[42], sometimes impairing between-group interactions. Consequently, adult hyraxes may maintain their social associations constant throughout the day to cope with (1) heightened levels of sexual competition, (2) predation risk, (3) as a by-product of social continuity, or any combination of these factors, resulting in strong correlations between 'active' and 'passive' interactions during the day.

Rock hyraxes maintain a relatively stable social structure throughout the year, except during the mating season when bachelors occasionally interact with social groups[32]. Thus, we can predict that the structure of correlation between daytime and nighttime networks is even stronger before the mating season, especially in spring when groups communally raise their pups[30]. In the absence of sexual interactions or vulnerable pups in winter, individuals tend to form weaker bonds, as documented in rhesus macaques (*Macaca mulatta*)[47], which may lead to more unpredictable network structure and weaker day-night correlations. Further investigation on how daytime and nighttime social networks influence each other should be carried out on semi-underground species, during and outside periods of sexual activity, for instance via experimental manipulation of den access.

Hyraxes forage outside their den during the day and are therefore exposed to predators. At night, although the risk of predation is negligible underground, the risk of intra-specific aggression may increase. The cost of this aggression is higher at night compared to daytime, as limited space underground forces

proximity, induces social stress, and may affect sleep quality. In free-moving mice, for instance, subordinates have shorter deep sleep stages than dominant individuals, due to differences in the costs of intra-specific aggressions[7]. In Japanese macaques, familiar individuals sleep better and longer than individuals sleeping with non-native conspecifics[16]. Hence, hyraxes are exposed to predation pressure during the day and intra-specific aggression at night, both of which are commonly associated with differentiated social relationships in animal networks. On one hand, under high daytime predation risk, differentiated relationships allow individuals to select social affiliates that are effective in deterring predators. For instance, ungulate species form more modular networks[48] and Trinidadian guppies become more assortative[49] and more selective[50] in high-risk environments. On the other hand, in contexts where spatial avoidance is not an option and the cost of aggression is high (e.g. inside dens), differentiated relationships provide support against intra-specific aggression and reduce social stress[51,52]. For instance, non-related spider monkeys maintain greater inter-individual distances while sleeping at night than related pairs[53], and tufted capuchins sleep closer to matrilineal kin than to unrelated individuals[17]. Consequently, differentiated social bonds are generally expected in groups of rock hyraxes, and, as the fear of being preyed upon usually outweighs the fear of intra-specific aggression, we expected a stronger social differentiation during the day compared to nighttime. Yet, we found that hyraxes are more selective at the individual level at night, although group-level social differentiation was not significantly different between daytime and nighttime. Additionally, the composition of hyrax social environment is almost constant over time, suggesting a stable group composition across daytime and nighttime. Together these results imply that hyraxes re-allocate their social interactions towards a few preferred individuals at night without changing group composition, which challenges our prediction that daytime predation risk triggers more differentiated social bonds during the day.

These results may find an explanation in the presence of an important mitigating factor: during the day, hyraxes forage

together under the surveillance of a sentinel constantly scanning their immediate surroundings[38,54], a behaviour commonly observed in socially cohesive or cooperatively breeding species[55,56]. In such groups, information on predators' presence is easier to acquire. Therefore, individuals rely more on group-level cooperation than on a few preferred affiliates to detect and/or deter predators, which reduces social differentiation[57]. In the rock hyrax, sentinel behaviour combined with a rocky environment rich in hiding spots[58] was proven highly effective—adult hyraxes are only rarely preyed upon by terrestrial predators[59]. Hence, the effect of predation risk on their social behaviour during the day is lower than expected while no apparent mitigating factors for nighttime social stress are at play. The combination of low predation risk under cooperative anti-predator behaviour during the day and social stress related to social sleeping in a limited space at night therefore drives more differentiated social bonds at night compared to daytime.

Accordingly, populations of hyraxes under higher daytime predation risk should display higher levels of social differentiation during the day than at night, or at least a smaller difference between daytime and nighttime social differentiation compared to our study population. Leopards were the rock hyrax's main terrestrial predator before going extinct in our study area over a decade ago. It is thus likely that our study population used to display different patterns in day/night social differentiation in the past. As for periods beyond hyrax annual mating season, it is uncertain whether patterns in social differentiation will persist. Sexual competition likely affects daytime and nighttime networks differently since bachelor males do not have access to groups' dens at night regardless of the season. Even though we can predict whether daytime social structure will be carried over during the mating season[60], quantitative changes in daytime and nighttime structures could lead to unpredictable patterns in the contrast between them. For instance, intra-specific competition for mating opportunities generally promotes higher intra-specific aggression rates[61]. It raises the question of nighttime forced proximity inside hyrax dens but also daytime avoidance of potential agonistic interactions with competing males. On one hand, decreased aggression risk inside the dens can lower the social differentiation at night, hence decreasing day/night variations in social differentiation overall. On the other hand, since intra-specific aggression mainly occurs between male hyraxes[62] and bachelor males having no access to groups' dens, we can assume most aggression occurs during the day. Thus, lower aggression rates during the day likely decrease daytime social differentiation and increase the contrast between daytime and nighttime networks as a result. Comparing daytime and nighttime differences in social structure between populations exposed to varying levels of predation and sexual competition could shed light on how wild animals use short-term network dynamics to cope with their environment while placing their daytime social structure in its ecological context.

Our results suggest that nighttime 'active' sociality is a favourable time when hyraxes can interact under minimal external pressures. Group-level standard deviations in eigenvector and strength centrality were significantly lower at night compared to daytime. These results suggest that group members converge towards more similar social behaviours at night, resulting in socially less diverse groups. Hyraxes must express a wide range of social behaviours to mitigate predation risk, improve food intake, and increase reproductive success during the day, driving animals to adopt different social niches[63]. Hence, group members are more socially different from one another during the day compared to nighttime. At night, on the other hand, hyraxes are under negligible predation risk, low thermal stress, do not need to forage, and free of sexual attention from competing bachelor males which do not have access to the social den. Thus group members have more time to socialise than during the day[64,65].

Dedicated periods of social interactions exist in multiple animal species, for example, the 'morning dance' of Arabian babblers[66], greeting rituals in mammals living in fission-fusion societies[67,68] or post-feeding sociality in Barbary macaques[69]. Observation-based studies revealed that interactions can influence group-level social dynamics across different social contexts[70–72]. For instance, allo-grooming networks predict agonistic support in non-human primates[73] and subordinates groom dominant individuals to reduce aggression rates in both meerkats[74] and Norway rats[75]. As 'active' nighttime sociality in hyraxes is only constrained by space use, nighttime social interactions are a better proxy for social preferences between-group members, and likely affect hyrax sociality in other social contexts. Thus, we propose that hyrax 'active' nighttime sociality serves a social function such as described in other species (e.g. social bonds maintenance, aggression reduction, etc.). Investigating context-dependent social structure in this species could thus uncover new aspects of hyrax social dynamics. In general, future studies combining the resolution of biologging devices with behaviour classification methods—such as accelerometers—could shed light on short-term social dynamics in wild species, significantly advancing our understanding of the ecology of group-living animals.

Overall, our work revealed a persistent topology along unstable edge weights at multiple timescales in hyrax networks. At night, hyraxes maintain the structure of their binary network (i.e. high stability of neighbours, constant degree centrality, constant eigenvector centrality, and network density) while being less social (i.e. lower strength centrality), and more selective of their affiliates (i.e. higher individual selectivity). These results suggest that rock hyraxes do not rewire their network between daytime and nighttime, but rather redistribute their social interactions within the same group of individuals. The same result repeats itself at a monthly scale with binary networks being more correlated than expected by chance whereas weighted networks do not display such a pattern. Thus, hyraxes may actively maintain existing social bonds over time (e.g. monthly scale) while using differentiated relationships to navigate rapidly changing socioecological contexts at shorter timescales (e.g. day vs. night ecological conditions).

Several studies showed that animals maintain a stable social structure across years while displaying variability between seasons[47,76–81]. Such patterns improve individual fitness through the establishment of long-lasting and valuable social bonds[82,83] while still allowing groups to respond to predictable changes in their physical environment[84–86]. For example, a seasonal decrease in food availability promotes networks of lower density where group members interact in smaller clusters to decrease intragroup competition for food[87]. This dual aspect of social relationships bears an adaptive value and must therefore be subjected to selective pressures. However, few studies investigated such social dynamics over periods shorter than a season and we have little understanding of how complex dynamics emerge at large topological and spatiotemporal scales from very short-term patterns. Additionally, most theories on social stability are based on non-human primates or species known for their complex social structure[81,88–91]. This bias may erroneously suggest that social complexity (see[92,93]) is a requirement to establish social relationships with this dual nature. But just like complex movement coordination is achieved in bird flocks and fish schools[94,95], complex network dynamics can be observed in social species that do not necessarily display complex multilevel social behaviours, and long-term population-level social stability can emerge as a by-product of simple daily social tactics.

Studying short-term network dynamics carries technical challenges for behavioural ecologists. Traditional approaches in animal networks rely on the repeatability of social bonds over time to accurately infer group social structure: the more a connection repeats itself, the more likely it is to represent a strong social bond. To increase the repeatability of observed social contacts—and therefore the reliability of the inferred network—researchers usually aggregate interactions over long periods, which may be the reason behind the apparent lack of studies on very short-term dynamics in animal networks. When investigating the structural changes occurring over very short periods such as day-night changes, aggregating networks over long periods makes little sense as the gain of repeatability is not on the structure itself but on its transitions.

In this work, we aggregated social networks at the scale of the day to increase the number of observations of transitions from daytime to nighttime structures. Our framework thus comes with technical caveats which are important to address. First, our approach considers rare interactions with the same importance as routinely repeated social contacts. Therefore, some daily networks may not accurately represent the regular social behaviour expressed by a study population over longer periods, which imposes limitations on conclusions that can be drawn from such networks. This method also comes at the cost of daily networks being sparse, which calls for careful methodological considerations when analyzing data (e.g. excluding sensitive networks traits, excluding empty time periods) and interpreting the results (e.g. loss of statistical power in networks built on few interactions). For instance, we detected a significant individual-level social differentiation (i.e. individual selectivity) whereas group-level social differentiation was non-significant according to the permutation test. This pattern is likely due to network sparsity: sparse networks include a relatively large amounts of zeros, which significantly lower the social differentiation score of individuals or groups—despite null models controlling for structural zeros in the network. Some group members have low daily degree centrality although they interact with all group members across several days (i.e. they interact with few group members each day, but change their social partners regularly). Consequently, they impose many zeros in the group structure at the daily scale which would not be the case if aggregating contacts over longer periods. Such local sparsity results in small differences between daytime and nighttime group-level social differentiation, low statistical power, and ultimately a non-significant permutation test.

Statistical tools capable of handling network sparsity were developed by network scientists[96,97]. Yet few of them were successfully transferred to behavioural ecologists or are adapted to the sample sizes usually encountered in behavioural ecology. Increasing the transfer of knowledge between network science and the study of animal behaviour will provide new tools to track very short-term network dynamics in social systems and may significantly advance our understanding of social stability across timescales. It may also help explain presumably altruistic behaviours such as the evolution of cooperation among kin and non-kin, aided by insights about the constraints imposed by the network structure at multiple temporal scales[98].

## Materials and methods

**Data collection and sampling**. We conducted fieldwork in the Ein Gedi Nature Reserve in Israel (31° 28′ N, 35° 24′ E) on two distinct study sites located approximately 2.5 km apart. Hyraxes were trapped between March and June 2017 to be marked according to previously published protocols[99]. Any trapped hyrax heavier than 1.8 kg was anaesthetised using 0.1 mg/kg of ketamine hydrochloride (intramuscular injection) and fitted with a Sirtrack E2C-171-A proximity biologger. Out of 83 individuals present in our study area, 37 were old and heavy enough to receive a biologger and 28 (14 females, 14 males) were successfully equipped for a

period of 27 consecutive days between July and August 2017. The study period covered hyraxes' annual mating season in its entirety. Having assessed loggers' quality under laboratory conditions[100] before deployment, we deployed in the same area loggers that consistently performed well together, whereas pairs of loggers showing poor performance were deployed in different study sites to minimise their chances of encounter. Hyraxes were trapped again at the end of the field season to retrieve their collars (see Supplementary Methods). Notably, 7 proximity loggers were either never retrieved, or permanently damaged, resulting in the loss of the data they recorded.

**Ethics approval**. Handling protocols for hyraxes in Ein Gedi Nature Reserve were approved by the Israeli Nature and Parks Authority (Permit number: 2017/41507). Efforts were made to reduce animals' stress and handling time.

**Constructing proximity-based networks**. Due to inter- and intra-logger variability[101], proximity data require multiple corrections to obtain reliable lists of social interactions (see Supplementary Methods).

Each pair of proximity loggers normally store duplicated records of their encounter in their internal memory. Because some loggers were never retrieved, the social behaviour of their carriers was only recorded by other devices. To correct for missing collars, we removed duplicate proximity contacts from dyads where both collars were retrieved by randomly excluding the records from one of the loggers every day[28]. We then repeated the data analysis multiple times to ensure that our results were qualitatively robust to the subset of loggers retained by this random selection.

As part of raw data pre-processing, we divided the study period into intervals of five minutes for which each dyad received a value of either 0 (no interaction during the interval) or 1 (the dyad did interact during the interval). A 5-min interval when a dyad is found interacting is considered a 'proximity event'. After pre-processing, the dataset consisted of 15,047 proximity events.

Proximity events were aggregated in various manners in this study to highlight variations in hyrax social structure over time. For each period of aggregation, we built weighted social networks using the simple ratio index[102] in the 'asnipe' R package[103]. Patterns of aggregation, subsequent data analysis, and methodological considerations leading to these choices are described below.

**Permutation tests**. Because social networks violate the basic assumption of data independence, classic statistical tests are likely to return significant results even when the social structure is random. In animal behaviour such patterns can be a by-product of group composition or ecological biases. For instance, non-random animal space use increases the chances of encounter between individuals occupying the same space despite an absence of social attraction between them. It is thus a common practice to compare observed network traits with traits that would be observed while accounting for these confounds using permutations.

In this study, we work under the null hypothesis that hyrax sociality is the result of random associations between individuals sharing a common territory at the same time. For every network presented in this study, we performed 1000 focal data-stream permutations[104], designed to account for space use bias while simulating random associations between individuals in the same area. As we do not have direct access to animal space use data, we used a community-detection algorithm to approximate groups of individuals that were more likely to interact by chance. Permutations were thus not restricted within physical territories, but within groups as detected by the community-detection algorithm. We used the Overlapping Cluster Detection algorithm from the 'linkcomm' R package[105], run on a weighted network based on the full dataset of 15,047 proximity events (Fig. 1), since it provides a better approximation of animal likelihood of encounter than other algorithms without overlapping communities.

As some individuals interact with multiple social groups every day, they can be assigned to several communities at the same time. For these individuals, we defined their 'group' as the union of all their communities and their social interactions could be permuted within these groups to simulate random associations. Preliminary exploration of proximity data revealed that only 7 proximity events (0.04%) occurred between the assigned communities, suggesting the algorithm correctly detected group structures. We further retain this definition of groups when investigating temporal patterns in individual and group-level network traits.

Data-stream permutations were restricted within communities and within days to account for temporal structure in the raw data. In other words, we did not swap contacts occurring in two different days (or communities) as it would no longer control for the temporal correlation between days (or the spatial correlation between individuals of the same community). The permutations produced 1000 random networks which structures represent the null hypothesis that hyrax spatial distribution is the main driver of their interaction patterns every day. When calculating a network trait on an observed network, we compared it to the distribution of 1000 random estimates obtained from the permuted networks. If the observed network trait fell within the lower (or upper) 5% of the random distribution, we concluded that our null hypothesis did not explain the observed data, and the network was influenced by an alternative social process. We summarise this information under the form of permutation-based $p$ values

calculated as the number of times the observed values are larger than their random estimates, divided by the number of permutations.

Unless another statistical test is explicitly cited, all reported *p* values were calculated using the above permutation procedure. All analyses were performed in R version 4.0.1[106].

**Preliminary exploration of the data**. We estimated the average number and average duration of social contacts between daytime and nighttime. As there was no significant temporal autocorrelation in the time series of the number of proximity events per day and per night (i.e. non-significant port-manteau test from the '*stats*' R package[107]), we directly compared the number of social contacts recorded between consecutive days and nights using a student t-test for dependent samples. As the time series of average social contact length per day and per night was slightly autocorrelated, we compared the average length of social interactions between day and night using a classic Wilcoxon rank test for dependent samples and ensured the *p* value was robust to the time dependence within samples using permutations for paired samples.

**Discriminating between 'passive' and 'active' sociality at night**. We built a series of networks based on nighttime interactions of different lengths and analyzed the structure of correlation between them. If a threshold exists in interaction length after which hyraxes express 'passive' social encounters (as opposed to 'active'), we should observe a sudden drop in average strength centrality and cosine similarity indexes as the proportion of long social encounters in the time-aggregated network increases (Supplementary Methods). We used the cosine similarity index implemented in the '*lsa*' R package[107]. It is the dot product between two sequences of numbers, divided by the product of their lengths[108], and measures the cosine of the angle between two numerical vectors when projected in an inner product space. Its interpretation is comparable to classic correlation coefficients such as Pearson's R and it has the advantage not to depend on vectors' amplitude or size, but only on the angle between them.

**Social structure across phases of the day and social contexts**. Considering the fundamental differences in behavioural states and social contexts when animals are resting compared to when they are active, and because ecological conditions are radically different between daytime and nighttime, we aggregated social interactions into 'daytime' and 'nighttime' networks and further divided them into 'passive' and 'active' sub-networks. We then compared these four network types using the cosine similarity index.

Observed cosine similarity indexes were directly compared to the random distribution of cosine indexes obtained via 1000 data-stream permutations and *p* values were extracted as a proxy for a significant deviation from our null hypothesis (see *Permutation tests*). As we compared each social network to multiple other networks, we increased the Type I error rate. Hence, permutation-based *p* values were adjusted using the False Discovery Rate (FDR)[109] implemented in the '*stats*' R package[106]. The FDR protocol specifically controls for increased Type I error rate when performing multiple comparisons under null hypothesis testing[109].

**Comparing 'active' social networks across days**. We studied how the 'active' social structure changes over time by dividing the proximity contacts into 54 distinct time periods representing the days and the nights of the study period (27 days). We filtered out 'passive' proximity events and aggregated the remaining 'active' proximity events into 54 sparse time-aggregated networks. Each one of the 54 daily networks was built by aggregating 131.8 (±sd = 32.6) proximity events on average (see Supplementary Methods on daytime/nighttime definitions). Nighttime networks were built from shorter times periods (99.6 ± 2.7 events) than daytime networks (163.9 ± 2.2 events). After filtering out the 'passive' proximity events, we detected 97.8 (±33.4) 'active' proximity events per aggregation period on average. These results imply that hyraxes do maintain some level of activity during the night, although the predominance of sleep causes individuals to display lower strength centrality at night compared to daytime.

We calculated the cosine similarity index between every possible pair of 'active' networks, resulting in a 54 × 54 matrix of cosine similarity indexes. We repeated this analysis on both weighted and binary networks and compared the average cosine similarity indexes between weighted and binary correlation matrices using a permutation test for paired samples. Because daily social networks are relatively sparse, structural zeros artificially increase the observed cosine similarity indexes. Therefore, all cosine indexes reported here only make sense when compared to each other via the data-stream permutation pipeline described earlier, while isolated raw cosine indexes are less informative due to matrices' sparsity. Hence, observed cosine similarity indexes were directly compared to the distribution of 1000 random cosine indexes obtained via data-stream permutations (see *Permutation tests*). Again, as we compare each social networks to multiple other networks, permutation-based *p* values were adjusted using the False Discovery Rate[109] implemented in the '*stats*' R package[106].

**Comparing 'active' network traits between day and night**. We then characterised the social structure emerging at night compared to daytime. To do so, we calculated 5 node-level and 2 group-level network traits of interest on the 54 time-aggregated networks described above using the '*igraph*' R package[110]. For group-level traits, social groups were defined using the Overlapping Cluster Detection algorithm from the '*linkcomm*' R package[105].

Denser networks were reported in animal societies experiencing ecological stressors. Thus, we predicted that daytime exposure to predation risk would trigger denser networks during the day in hyraxes. Hence we calculated the density of edges—defined as the number of existing ties divided by the number of potential connections—for every day and every night of the study period. We then calculated the individual number of direct connections in the network (i.e. degree centrality) and the individual sum of edge weights (i.e. strength centrality) which are indicators of how social an individual is. Because of considerable changes in predation risk, foraging activities, and physiological states between day and night, we expected hyraxes to be better connected to all other group members during the day as central individuals have better access to social information and better chances to survive predation events. Hence we considered the individual eigenvector centrality, which accounts for how well a node is connected to the rest of the network, based on its direct connections and the connections of its neighbours. Aggregating social contacts over short time-period generates relatively sparse networks, hence we excluded centrality measures deemed too sensitive to network sparsity (e.g. betweenness centrality, closeness centrality). We calculated the coefficient of variation (CV) in edge weights at the individual level (i.e. individual selectivity) and at the group level (i.e. social differentiation). This measure indicates how variable edge weights are, providing a proxy for animal tendency to favour some relationships over others. High CVs indicate differentiated relationships at the individual level and strong choosiness in social partners at the group level. More differentiated social bonds were reported under predation risk; hence we expected hyraxes to display higher CVs during the day compared to nighttime. Finally, we evaluated the stability of hyraxes' social affiliates (i.e. neighbours' stability) by calculating the Jaccard index on a node's neighbours between consecutive periods of aggregation (Supplementary Methods). We also calculated the standard deviation of 3 individual network traits within groups (i.e. degree centrality, eigenvector centrality, and strength centrality), since group homogeneity in social behaviour promote individual survival in rock hyraxes[31].

We calculated daily absolute differences in networks traits between consecutive periods of aggregations (e.g. night of date n, and day of date n + 1) and compared them to random estimates obtained via 1000 data-stream permutations. The permutation tests returned a series of dependent uncorrected *p* values calculated as the number of times the observed absolute difference was larger than its corresponding random estimate. We then combined these *p* values into one overall *p* value per network trait using the competitive test for dependent samples from the '*CombinePValue*' R package[111]. We considered *p* values smaller than 0.05 to be a sign of deviation from our null hypothesis—meaning that differences between daytime and nighttime network traits were significantly larger than predicted by hyrax space use.

**Statistics and reproducibility**. We reported sample sizes and relevant statistical parameters in the 'Material and methods' and 'Results' sections. Further details on population size and composition as well as loggers' coverage are included in Supplementary Table 1 and Supplementary Table 2. All statistical analyses were performed in R (v4.0.1).

**Reporting summary**. Further information on research design is available in the Nature Portfolio Reporting Summary linked to this article.

## Data availability

All data and intermediate datasets generated in the process of this study are available on a Zenodo repository at the address https://zenodo.org/badge/latestdoi/388145736. Two files exceeding repositories' size limit will be shared upon request.

## Code availability

R codes used to analyse the data and to produce the figures are available on a Zenodo repository at https://zenodo.org/badge/latestdoi/388145736 and on the author's Github page at camillebordes/Hyrax_daily_SND.

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

## Acknowledgements

We thank Nikki Thie for her help deploying proximity loggers in the field and Alexander Christensen who provided guidance to analyze temporally correlated paired samples of social network traits. This work was supported by the Israel Science Foundation (550/14, 767/16, 244/19, 245/19), the US-Israel Binational Science Foundation (2015088, 2019156), and a grant from Bar-Ilan University Data Science Institute. We also thank the Ein Gedi Nature Reserve for their support in our daily activities.

## Author contributions

C.B. and A.I. designed the experiment. C.B., R.B., and Y.G. collected the data. C.B. designed and performed the analysis. C.B. led the writing of the paper. A.I. and L.K. provided guidance. All authors participated in the revision of the paper.

## Competing interests

The authors declare no competing interests.
