## [Peer Review File · Communications Biology]

Reviewers' comments:

Reviewer #1 (Remarks to the Author):

See attached file.

Reviewer #2 (Remarks to the Author):

"High-resolution tracking of social interactions highlights nocturnal drivers of animal sociality" is well-done manuscript that highlights hard to observe social patterns. This is a very interesting topic. The manuscript, however, has some points that should be clarified and potentially updated. Major comments:

1. Certain terms seem to be overstatements or conflated given what is actually being tested in the paper. Firstly, "optimal" sociality is discussed throughout the manuscript, but what optimal is defined as or represents biologically doesn't seem to be discussed. Furthermore, sleep, nighttime, and nocturnal are conflated throughout the paper. It would be helpful to go through the paper and come up with consistent terminology. For example, lines 67-68 mention "nighttime ecology" and "main activity periods"; nocturnal animals' main activity periods are at night, but this is implying main activity periods are during the day.
2. Certain aspects of the methods need more clarity to be understood and replicated.
 - a. A lot of the social network metrics don't have biological reasoning for their use. I'm sure there are reasons! But it would be good to spell it out since SNA is just a tool to describe biological processes. This should be done for each network metric. This will also help people who don't know SNA to understand what is being discussed.
 - b. What actually is a cosine similarity index, what does it do? I'll admit I'm unfamiliar with this. The False Discovery Rate protocol should also be further explained
 - c. When you say permutations are restricted within groups and time periods, do you mean the 5 minute time periods, active/passive time periods, or something else?
 - d. Certain tests are mentioned in the results that aren't in the methods such as t-test, Wilcoxon test. And explain why these tests are appropriate for this non-independent data.
3. In the results, given the data shown, I'm surprised at what is and what is not significantly different. In the violin plots, it might help to plot the raw data as well (Figs. 3 &4).
4. The assertion that nighttime networks drive daytime networks is a bit of a circular argument. It could be that their daytime associations drive their nighttime ones because of their spatial assortment. Without some sort of causal analysis, this claim seems to be an overstatement. It's definitely worth discussing! But I think the paper would be stronger with more discussion of the caveats of the data, analyses, and results.
5. Building on the discussion of caveats from the previous comment, the fact that this data is from the mating season seems like a throw-away comment at the end. I would think that the mating season would have a significant impact on social structure, so representing these results as what hyraxes do all the time may be misleading. I would say from the beginning make it clear the mating season social structure is what is being studied and be more nuanced throughout the paper about what the data actually represent about hyrax social structure. These findings aren't really general, but specific to an important period in the hyrax lifecycle.

Minor comments:

Line 56: It's been long enough since you said what the "two factors" are, so I would restate them here.

Lines 86-87: I would briefly define structural balance.

Lines 113-121: This doesn't really belong in the Intro, but it would be a good conclusion.

Line 150: the period after "But" should be a comma

Lines 503-507: The discussion of the importance of looking at shorter time scales is a bit weak. What knowledge do we gain at shorter timeframes than seasons? Usually researchers seem to want longer timeframes to be more representative. The seasonal flexibility is a very important

point, but this paper doesn't really address that either (related to my major comment #5). I would either further discuss these points and how it relates to the caveats of the study or remove this section.

The following study compares day and night social networks for rock hyraxes through time and poses some interesting questions about night time ecology. I liked the paper, I think the analyses were well done and clever, and I think it was well written (a few notes below on style though). I think my comments are mostly semantic in that I think the authors use language that makes the study seem more novel than maybe it is? I am very familiar with the social network literature and while I agree there are almost no studies that explicitly quantify and test hypotheses about night time ecology, I think night time social behaviour of animals is implicit in many social network studies that the authors ignore. I also think there are some other points in the paper that the authors should avoid using statements of novelty and consider that perhaps there is other similar work that may not be as explicit, but certainly is implicit. I've provided a number of suggested references, but I'm sure I've missed lots of papers too.

Major comments:

Assertion that we don't study animals at night: at lines 67-68 the authors state "Despite the importance of nighttime ecology little attention has been given to animal sociality outside their main activity periods". I disagree with this. What about bats? Or any other animal where network edges are generated based on animals sharing a den/roost/nest/burrow or any animal with GPS/proximity devices that collect data throughout the day. I think there is a point to be made that in these cases we usually don't explicitly talk about (or parse out) nighttime ecology – but I disagree that "little attention has been given to animal sociality outside their main activity periods". I would suggest the authors rephrase to frame this statement to specifically be about diurnal species or to be about primates or to rephrase and qualify with some of the information I mention above.

At line 78 there are a few exceptions presented... but I would argue there are so many more papers out there and in fact these are not exceptions but actually represent a fairly large number of papers that either quantify networks throughout the day and night (and don't discriminate between the two). Here are just a few examples I found in a very quick search through my reference manager:

Zeus et al. 2017 (<https://doi.org/10.1016/j.anbehav.2016.11.015>): Bats with PIT-tags roost (and presumably sleep) together during the day.

Wey et al. 2013 (<https://doi.org/10.1016/j.anbehav.2013.03.035>): Degus that sleep in burrows together.

Hirsch et al 2013 (<https://doi.org/10.1016/j.anbehav.2012.12.011>) : raccoons with proximity collars that monitor throughout the day/night.

Robitaille et al. 2021 (<https://doi.org/10.1093/cz/zoaa052>): caribou with GPS collars that monitor throughout the day/night.

I think there is a nuance here that the authors need to communicate. From my perspective, that is that: 1) there are many studies that quantify social networks using data pooled from day and night without discriminating; 2) there are fewer studies that quantify social networks for day/night separately; 3) there are even fewer studies that quantify night time social networks with a specific aim to test hypotheses about night-time ecology.

Time-aggregated networks (lines 221-239): I think I know what the authors did here – so I am going to write out my interpretation with the expectation that whether I am right or wrong, the authors clarify this section to make it clear what they did. So – my interpretation is that the authors generated networks for each night/day on each day of the study ($n = 54$ time-aggregated networks, as indicated on line 236). I see nothing inherently wrong with this approach. However, I think by separating the data into so many small units (i.e. days instead of weeks or months), the authors run the risk of comparing very sparse networks. Perhaps some summary information on these networks will be important for the reader to distinguish the strength of this “mini-networks”. Specifically, each network is a very narrow snapshot (i.e. 12 hours) into the social lives of the group. I would suggest the authors include some caveats associated with this analysis in the methods. For example, how does having such a short temporal window affect the network? Especially given the proximity data is divided up into 5 minute chunks (so, in a 12 hour period, that means there are $5 \times 12 = 60$ mins/hr $\times 12$ hrs/night/day = 144). That is, there are 144 observational units per network where an individual could have interacted with another individual. I think these kinds of details should be present in the methods.

Minor comments:

Lines 14-15: perhaps this is true for species where we typically conduct focal observations to quantify social interactions, e.g. primates. But not so true for animals with biologging devices that record throughout the day or for animals that we generate measures of social structure based on patterns of co-roosting/co-denning/co-nesting at night. For example, almost every study on bats that uses social network analysis would generate association patterns of bats in the same roost during the day (i.e. when they are asleep).

Line 18: 27 consecutive days and nights? Perhaps work adding mention that nights were also monitored, since that is the main point of the paper.

Line 20: I think something is missing in this sentence? Better connect to the rest of their network.... At night? During the day?

Line 21: the pronoun “this” is confusing here – what does “this” refer to?

Line 22: which temporal scales?

Line 24: again, as above, the pronoun “this” is confusing – what is “this”?

Line 28: again, as above, the pronoun “this” is unnecessary... perhaps “our work” is more appropriate.

Lines 36-38: Are there references for these statements? I believe that social sleeping would certainly improve predator detection and benefit social thermoregulation.... But I am curious if there is a reference to indicate that social sleeping improves sleep quality?

Line 354: I think the use of “your” here is informal – is there a way to rephrase to refer directly to the individual hyraxes as opposed to “your social group”.

Lines 368-372: This section really caught by off guard – I think the authors need to indicate in the introduction or methods that their study takes place during the mating season. Arguably one of the most socially relevant seasons for animals! I was really caught off guard here... is there a way to integrate this more into the paper? Perhaps this is a missed opportunity. This also leads me to wonder of the 27 individuals with proximity devices, how many were males and how many were females?? This is REALLY important given the information that the study took place during the mating season! Please provide more information about this.

Line 429: I think the use of “interesting” is a bit informal and one could argue that any suggestion for a follow up study could be “interesting”... perhaps consider rephrasing to something more objective.

Line 448: I’m not sure I would invoke optimality here. I think that networks are so dynamic and fluid through space and time that I think it is almost impossible to truly quantify an optimal social network. Moreover, I think things like fitness, energetic status, foraging, etc need to be incorporated somehow and in this study we only have discussion of these factors – so even if the networks measured here are indeed “optimal”, there is no way for the reader to make a judgement without more data on things like fitness, energetic status, or foraging. I would therefore suggest removing (or significantly decreasing the length) this paragraph of the discussion.

Lines 503-506: I’m not sure I agree with this... many social network papers examine seasonal differences and/or stability/consistency over time. I did a quick google scholar search and came up with these articles:

Hobson et al. 2013: <https://doi.org/10.1016/j.anbehav.2012.10.010>

Jones et al. 2019: <https://doi.org/10.1002/ece3.5060>

Cantor et al. 2020: <https://doi.org/10.1016/j.anbehav.2012.06.019>

And these are articles I am more familiar with and have read/cite in the past:

Firth and Sheldon 2016: <https://doi.org/10.1111/ele.12669>

Robitaille et al. 2021: <https://doi.org/10.1093/cz/zoaa052>

Prehn et al. 2019: <https://doi.org/10.1016/j.anbehav.2019.08.018>

And I am sure there are many more. Anyways, my point is that I think the claim that there are no studies that examine the mechanism by which seasonal flexibility is achieved but long-term stability is maintained is misleading.

Lines 518-522: I think there may be a stronger way to conclude the paper than suggesting that novel technology will allow future studies to investigate social dynamics. Perhaps the authors could link their work to broader theory on the causes and consequences of social behaviour?

Figures:

Figure 1: I like this figure, but I think it is missing some information. It isn't clear which community a given individual is part of because of the pie chart in each node. Can there be some sort of kernel or outline around the individuals that were assigned to a given community AND the pie chart in each node? Otherwise, I would suggest removing the pie chart and simply coloring nodes by community... as it stands currently I find it confusing to tell which individual is in which community when they interacted with individuals from other communities. It might also be good to indicate how many communities are in the network... some of the colours presented look very similar and for anyone who is colour blind, the nodes in this network are going to be almost indistinguishable. Also, I think line 182 should read that the pie charts within each node represent the proportion of social interactions by that individual allocated to each community.

Figure 5: I like how this figure displays a very large volume of data – I think it is a really clever way to display this type of information. However, I also find it pretty overwhelming to get a meaningful idea of what's going on. For example, in the lower right panel, there was clearly one day where all of the networks were either lower than random (blue) or non-significant. What is going on there? Maybe a solution would be to have one very large table (or spreadsheet?) in the appendix with these data presented and perhaps a brief explanation of any anomalies?

REVIEWERS' COMMENTS:

Reviewer #2 (Remarks to the Author):

I was happy to see "High-resolution tracking of social interactions highlights nighttime drivers of animal sociality" again. The authors have made dramatic changes to the manuscript and greatly improved it. At this point, my comments would mainly be personal preferences, I will not include them here.

My only remaining comment would be to discuss the fact that these networks are mating season networks in the Discussion as well and relate it to the broader literature. Would you expect these patterns to hold outside of the mating season? How does this relate to other species with a defined mating season?

REVIEWERS' COMMENTS:

I was happy to see "High-resolution tracking of social interactions highlights nighttime drivers of animal sociality" again. The authors have made dramatic changes to the manuscript and greatly improved it. At this point, my comments would mainly be personal preferences, I will not include them here.

My only remaining comment would be to discuss the fact that these networks are mating season networks in the Discussion as well and relate it to the broader literature. Would you expect these patterns to hold outside of the mating season? How does this relate to other species with a defined mating season?

We extended the scope of the discussion to predictions beyond hyrax mating season as per Reviewer's 2 suggestion. Changes can be found in several places of the discussion:

"This result is unlikely to change outside the mating season as a similar daily routine was observed in the absence of sexual interactions." (lines 269-271)

"Rock hyraxes maintain a relatively stable social structure throughout the year, except during the mating season when bachelors occasionally interact with social groups³². Thus, we can predict that the structure of correlation between daytime and nighttime networks is even stronger before the mating season, especially in spring when groups communally raise their pups⁴⁸. In the absence of sexual interactions or vulnerable pups in winter, individuals tend to form weaker bonds, as documented in rhesus macaques (*Macaca mulatta*)⁴⁹, which may lead to more unpredictable network structure and weaker day-night correlations. Further investigation on how daytime and nighttime social networks influence each other should be carried out on semi-underground species, during and outside periods of sexual activity, for instance via experimental manipulation of den access." (lines 304-315).

"As for periods beyond hyrax annual mating season, it is uncertain whether patterns in social differentiation will persist. Sexual competition likely affects daytime and nighttime networks differently since bachelor males do not have access to groups' dens at night regardless of the season. Even though we can predict whether daytime social structure will be carried over during the mating season⁶², quantitative changes in daytime and nighttime structures could lead to unpredictable patterns in the contrast between them. For instance, intraspecific competition for mating opportunities generally promotes higher intraspecific aggression rates⁶³. It raises the question of nighttime forced proximity inside hyrax dens but also daytime avoidance of potential agonistic interactions with competing males. On one hand, decreased aggression risk inside the dens can lower the social differentiation at night, hence decreasing day/night variations in social differentiation overall. On the other hand, since intraspecific aggression mainly occurs between male hyraxes⁶⁴ and bachelor males having no access to groups' dens, we can assume most aggression occurs during the day. Thus, lower aggression rates during the day likely decrease daytime social differentiation and increase the contrast between daytime and nighttime networks as a result. Comparing daytime and nighttime differences in social structure between populations exposed to varying levels of predation and sexual competition could shed light on how wild animals use short-term network dynamics to cope with their environment while placing their daytime social structure in its ecological context." (lines 373-395)